Long-term follow-up of mandibular dental arch changes in patients with complete non-syndromic unilateral cleft lip, alveolus, and palate

Sumardi Sariesendy 1
Latief Benny S. 2
Kuijpers-Jagtman Anne Marie a.m.kuijpers-jagtman@umcg.nl 3 4 5
Ongkosuwito Edwin M. 6 7
Bronkhorst Ewald M. 7
Kuijpers Mette A.R. 6 7
1 Faculty of Dentistry, Department of Orthodontics, Universitas Indonesia , Jakarta , Indonesia
2 Faculty of Dentistry, Department of Oral-Maxillofacial Surgery, Universitas Indonesia , Jakarta , Indonesia
3 Faculty of Dentistry, Universitas Indonesia , Jakarta , Indonesia
4 Department of Orthodontics and Dentofacial Orthopedics, School of Dental Medicine, Medical Faculty, University of Bern , Bern , Switzerland
5 Department of Orthodontics, University of Groningen, University Medical Center Groningen , Groningen , Netherlands
6 Department of Dentistry - Orthodontics and Craniofacial Biology, Radboud University Medical Center , Nijmegen , The Netherlands
7 Radboud Institute for Health Sciences, Department of Dentistry, Radboud University Medical Center , Nijmegen , The Netherlands
Papageorgiou Spyridon
Electronic publication date: 2021 Dec 16
Publication date: 2021
Volume: 9
Electronic Location ID: e12643
Received 2021 Aug 10; Accepted 2021 Nov 26
Copyright: ©2021 Sumardi et al.
Copyright year: 2021
Copyright holder: Sumardi et al.
License: This is an open access article distributed under the terms of the Creative Commons Attribution License, which permits unrestricted use, distribution, reproduction and adaptation in any medium and for any purpose provided that it is properly attributed. For attribution, the original author(s), title, publication source (PeerJ) and either DOI or URL of the article must be cited.
License URL: https://creativecommons.org/licenses/by/4.0/

Keywords: Complete unilateral cleft lip and palate; Long-term outcome; Mandibular dental arch; Modified Huddart-Bodenham Index; Unilateral cleft lip, alveolus, and palate; Dental casts

Funding: Doctoral Degree Scholarship for Lecturers of Beasiswa Pendidikan Pascasarjana Dalam Negeri (BPPDN) Ministry of Education, Culture, Research and Technology, Republic of Indonesia Doctoral Candidate Grant of Universitas Indonesia This work was funded by a Doctoral Degree Scholarship for Lecturers of Beasiswa Pendidikan Pascasarjana Dalam Negeri (BPPDN), Ministry of Education, Culture, Research and Technology, Republic of Indonesia, and by a Doctoral Candidate Grant of Universitas Indonesia.

==============================
Background

Treatment of cleft lip and palate (CLP) requires a comprehensive interdisciplinary approach and long-term follow-up. Only a few studies are available that reported on changes after treatment, which showed that in particular the transverse dimension, in patients with CLP is prone to changes after treatment. However, those studies did not pay attention to concomitant changes in the mandibular arch that occur after treatment.

Objectives

To evaluate mandibular transverse dental arch dimensions and interarch transverse changes in patients with complete non-syndromic unilateral cleft lip, alveolus, and palate (CUCLAP) up to five years after treatment.

Material and Methods

Retrospective longitudinal study in 75 consecutive patients with CUCLAP directly after comprehensive treatment (T0), two (T2), and 5 years after treatment (T5). Great Ormond Street, London and Oslo (GOSLON) scores were available for all patients. Three-dimensional scans of all dental casts were made. Inter premolar and intermolar distances between the mandibular contralateral teeth were measured. The modified Huddart Bodenham (MHB index) was applied to assess the transverse interarch relationship. Paired t-tests and ANOVA were used to analyze transverse and interarch transverse changes. Linear regression analysis was done to define contributing factors.

Results

Paired t-tests showed a significant decrease of the mandibular inter first and second premolar distances (p < 0.05) and an increase of the inter second molar distance, whilst the MHB Index deteriorated at all time points for all segments and for the total arch score (p < 0.05). Linear regression showed no significant contributing factors on the decrease of the transverse distances. However, inter arch transverse relationship was significantly affected by age at the end of treatment, missing maxillary lateral incisor space closure, and the GOSLON Yardstick score at the end of treatment (p < 0.05), especially during the first two years after treatment.

Conclusions

Changes occurred in the mandibular arch expressed as changes in the transverse dimensions and interarch relationship measured by the MHB Index. A younger age at the end of treatment, space closure for a missing maxillary lateral incisor and a higher GOSLON score at the end of treatment negatively influence the interarch transverse deterioration especially in the first two years after treatment. For the transverse dimensional changes in the mandibular arch such influencing factors could not be determined.

Introduction

Unilateral cleft lip and palate is one of the most common birth defects. Most unilateral cleft lip, alveolus, and palate (UCLAP) are non-syndromic (Dixon et al., 2011) with a prevalence of 1.37 per 1,000 births in low- and middle-income countries (Kadir et al., 2017). Mossey & Modell (2012) estimated the prevalence per 1,000 births in South East Asia being1.08, Middle East 0.72, North America 1.17 and in Europe ranging from 0.63–1.07. Also, it has been found that the prevalence of cleft lip and palate (CLP) in males is higher than in females (Martelli et al., 2012; Yılmaz, Özbilen & Üstün, 2019).

It has already been recognized for a long time that treatment of these birth defects requires a comprehensive interdisciplinary approach. The American Cleft Palate-Craniofacial Association (2018) developed guidelines for team treatment for patients with orofacial clefts. Whilst the team provides comprehensive and integrated treatment, follow-up of the patients is also an integral part of the team duties. Following patients over a long period of time is not only important to assess treatment outcome within a team, but also to compare the results with other centers, in order to improve the care of patients with clefts.

Maxillary dental arch transverse dimensions and dental arch relationships following different treatment protocols for patients with orofacial clefts have been the topic of studies and systematic reviews that evaluated treatment outcome in relation to different treatment parameters such as timing of palatal surgery in relation to craniofacial outcome (Kappen et al., 2018), timing of primary alveolar bone grafting (Carbullido et al., 2021), outcome of maxillary distraction (Kloukos et al., 2018), and rehabilitation of the maxillary arch with a dental implant or an anterior fixed partial denture replacing the missing lateral incisor at the cleft side (Caballero et al., 2019; Rezende Pucciarelli et al., 2020; Pucciarelli et al., 2021; Soares et al., 2021a; Soares et al., 2021b). However, those studies did not pay much attention to concomitant changes in the mandibular arch of patients with cleft lip and palate that occur after treatment. To our knowledge no studies are available in the literature that describe transverse changes in the mandibular arch. It is well-known that the maxillary arch, and in particular the transverse dimension, in patients with a cleft palate is prone to changes after treatment (Marcusson & Paulin, 2004; Al-Gunaid et al., 2008; Semb, Ronning & Abyholm, 2011; Rezende Pucciarelli et al., 2020; Soares et al., 2021a; Soares et al., 2021b). Especially because often a considerable amount of expansion is needed during the orthodontic treatment to overcome the constrictive effect of palatal scar tissue. It may be possible that the mandibular arch will adapt, at least—partly, to post-treatment changes in the maxillary arch, but it may also be possible that the mandibular arch remains stable. This would result in disturbance of the transverse relationship between the arches.

Several factors could influence mandibular arch changes and interarch transverse relationship after treatment, such as sex and age at the end of treatment, orthodontic opening or closing of the space of a missing maxillary lateral incisor (Oosterkamp et al., 2010), extractions in the mandibular arch, expansion of the maxillary arch in relation to secondary bone grafting (Emodi et al., 2015), and interarch relationship as represented by the GOSLON (Great Ormond Street, London and Oslo) Yardstick at the end of treatment. These factors may predispose to changes in the dental arches for which the operator should anticipate with the orthodontic retention appliances.

The aim of this study was to determine long term changes in transverse mandibular arch dimensions and to determine interarch transverse relationship in patients with a complete non-syndromic unilateral cleft lip, alveolus, and palate (CUCLAP) until 5 years post-treatment. The null hypothesis was transverse mandibular arch dimensions and the transverse interarch relationship in patients with CUCLAP do not change until 5 years post orthodontic treatment. The risk factors that could affect the stability of the transverse mandibular arch dimensions and the transverse interarch relationship were also studied.

Materials & Methods

Subjects and treatment protocol

This is a retrospective longitudinal study in patients with a non-syndromic CUCLAP. Inclusion criteria were: non-syndromic as ascertained by the clinical geneticist, Caucasian, born between 1 January 1983 –31 December 1992, Simonart’s band allowed, consecutively treated from birth on with the same treatment protocol in the Cleft Lip and Palate Craniofacial Center, Radboud University Medical Center, Nijmegen, The Netherlands. Exclusion criteria were: additional craniofacial anomaly present, incomplete cleft, incomplete or submucous cleft lip at the other side, not Caucasian, primary operations performed elsewhere, treatment not yet finished.

All surgical procedures were performed by two experienced cleft surgeons and all patients were treated according to a standardized treatment protocol. The treatment protocol consisted of infant orthopedics followed by lip repair (Millard) at 4–8 months, soft palate repair (modified von Langenbeck) at 12–14 months, superiorly based pharyngeal flap at 4–5 years (if needed), maxillary protraction and expansion at 5–6 years for 12–18 months (if needed) followed by minor lip/ala nose correction (if needed), alignment of maxillary incisors and/or maxillary expansion at 9–11 years with a quad-helix or removable appliance followed by secondary bone grafting with a chin or rib graft together with hard palate closure (Boyne & Sands). The orthodontic treatment in the permanent dentition was performed with fixed appliances around the age of 12–15 years followed by fixed retention at the maxillary and mandibular anterior teeth and/or a Hawley retainer in the maxilla. Lastly, in case of severe dysgnathia, combined orthodontic-surgical treatment followed by lip/nose correction was performed, if indicated.

All patient data were anonymized prior to analysis. The use of anonymous data collected during routine patient care is in accordance with Dutch law. Written consent of participants was obtained. This study has been carried out in accordance with the applicable Dutch legislation such as Medical Research involving Human Subjects Act and Medical Treatment Contracts Act concerning reviewal by an accredited research ethics committee. This has been confirmed by the Medical Ethical Committee of the Radboudumc Nijmegen, The Netherlands (File number: 2021-13046).

Materials

Plaster models of 75 consecutive patients directly after (T0), 2 years after (T2) and five years after treatment (T5) were available. The study started (T0) after orthodontic treatment had finished, either after orthodontic treatment alone or after combined orthodontic surgical treatment in case of severe dysgnathia. All plaster casts were scanned with a 3Shape R500 3D Dental Laser scanner (3ShapeR©, Copenhagen, Denmark) using the high-resolution setting, producing a spatial resolution of 0.01 mm as specified by the manufacturer. The occlusal relationship of all models was checked against the intra-oral pictures taken at the same date.

Methods

The mandibular transverse dental arch dimensions were measured by one observer on the digital models at T0, T2 and T5 using the open-source software MeshLab 2016 (Visual Computing Lab ISTI - CNR, Pisa, Italy) (Cignoni et al., 2008). The mandibular inter-canine width was measured between the left and right cusp tips, the inter first and second premolar width between the left and right buccal cusp tips, and the inter-first and second molar width between the mesio-buccal cusp tips (Fig. 1). To determine the error of the method 20 randomly selected digital dental casts were remeasured by the same observer after a one-week interval.

Figure 1 Mandibular transverse dental arch dimensions.

Measurements of the mandibular transverse dental arch dimensions.

The transverse occlusal relationship was determined by applying the MHB-index (Modified Huddart-Bodenham index) to evaluate arch constriction for each tooth pair. This was done on plaster casts set out on tables in a quiet room by one observer. This index was first developed in 1972 (Huddart & Bodenham, 1972) then modified in 1997 (Heidbuchel & Kuijpers-Jagtman, 1997). For the scoring the dentition is divided into a labial segment, and a buccal cleft and non-cleft segment (Fig. 2). In the labial segment the lateral incisors are not scored as they are often missing or misplaced. In the buccal segments, canines, premolars, and first molars are scored. Each tooth pair can receive a score from −3 to +1 (Fig. 3). The total MHB score for a given model is calculated as the sum of the scored teeth, with a range of the score from −30 to +10 (Noverraz et al., 2015). To determine the intra- and interobserver reliability 50 randomly selected casts were scored twice with an interval of one week by two observers.

Figure 2 Maxillary arch segments used for the Modified Huddart-Bodenham scoring.

The segments of the maxillary arch on which the Modified Huddart-Bodenham score for interarch relationship is determined.

Figure 3 The Modified Huddart-Bodenham scoring of teeth pairs to determine the interarch relationship.

Statistical analysis

Descriptive statistics were performed for all parameters to show the distribution of data, such as means, standard deviations, minimum –maximum values, mean increments between T0 and T2, T2 and T5, and T0 and T5. The intra-observer performance for the transverse mandibular distances (continuous variables) was analyzed using three statistics. The reliability was tested with the Pearson correlation coefficient. Paired sample t-tests were applied to identify systematic differences between the first and second measurement. The duplicate measurement error (DME) was calculated as the SD of the difference between two observations divided by 2. Additionally, Bland-Altman plots were made for each variable. The duplo measurements for the intra- and interobserver reliability of the individual teeth scores of the MHB Index score were analyzed by weighted Kappa. Data was analyzed using paired t-tests and ANOVA. Linear regression was applied to test for the effect of age at the end of treatment, sex, opening or closing space of the maxillary lateral incisor at the cleft side, mandibular premolar extraction, maxillary expansion before bone grafting, and treatment outcome (GOSLON Yardstick score) at the end of treatment. The level of significance was set at P < 0.05.

Results

Sample

The total number of patients that met the inclusion criteria was n = 82. Data of 7 patients were missing because 3 moved elsewhere, for 3 patients all treatment records were missing, and 1 patient had no follow-up for unknown reasons. In total 75 consecutive patients were available for the study. The mean age at the end of treatment (T0) was 16.5 years (SD 2.3; range 12.9–22.5), two years after treatment (T2) 18.4 years (SD 2.2; range 14.6–22.7), and five years after treatment (T5) 20.9 years (SD 2.4; range 17.0–26.4). 27 patients were female (36%) and 48 patients were male (64%). 54 patients (72%) had the cleft on the left side and 21 patients (28%) on the right side. Maxillary expansion before bone grafting was done in 20 patients out of 75 (26.7%). In 54 patients (72%) the space for the missing maxillary lateral incisor at the cleft side was closed orthodontically, in 6 patients (8%) the maxillary lateral incisor was present at the cleft side and in 15 patients (20%) the space was opened. Absence of mandibular premolars was found in 19 patients (25.3%). The GOSLON score at the end of treatment was in 36 (48%) patients score 1, in 19 (25.3%) patients score 2, in 14 (18.7%) patients score 3, and score 4 was found in 4 (5.3%) patients. There were no patients with a score of 5 in this sample. Data for the GOSLON were missing of two patients.

Intra- and interobserver performance

The Pearson’s correlation coefficients for the intra-observer reliability of the transverse arch measurements were high, ranging from 0.932 to 0.996 (Table S1). The duplicate measurement error for the measurements varied from 0.19 to 0.54 mm. The mean differences between first and second measurements were very small and showed no significant differences (paired sample t-test). This indicated that intra-observer performance was high. This was also supported by the Bland Altman plots showing a high level of agreement between the first and second measurements (Figs. S1A–S1C).

For the MHB score the intra-observer performance was almost perfect shown by the weighted kappa of 0.93. The inter-observer reliability was also strong with a weighted kappa value of 0.84. This was also supported by the Bland Altman plots showing a high level of agreement between measurements (Figs. S2A–S2B).

Mandibular transverse dimensions

Table 1 shows means and SD of the five mandibular transverse dimensions at the three different time points and Table 2 shows the changes over the five-year study period. The mandibular intercanine distance did not change significantly over time. The interpremolar distances diminished significantly over the 5-year study period. The first premolar distance diminished 0.71 ± 1.38 mm (p = 0.001, 95% CI [0.29–1.12]), while the second premolar distance decreased 0.95 ± 1.74 mm (p < 0.001, 95% CI [0.47–1.43]). The inter first molar distance increased slighty over the 5-year follow-up period but this change was not statistically significant, while the inter second molar distance showed an increase of 0.73 ± 1.49 mm (p = 0.001, 95% CI [−1.13–−0.33]).

Table 1 Mean and standard deviation (SD) for mandibular intercanine, inter first-premolar, inter second-premolar, inter first-molar, inter second-molar distances after treatment (T0), two years after treatment (T2), and five years after treatment (T5) (mean, SD in mm).

Mandibular distances	Time
point	N	Mean	SD	Minimum	Maximum	
3 - 3	T0	75	26.18	1.93	21.21	30.34	
 	T2	65	26.17	2.16	21.02	30.78	
 	T5	57	26.16	2.00	21.30	30.41	
4 - 4	T0	58	33.20	2.09	28.49	37.93	
 	T2	50	32.58	2.27	27.79	38.07	
 	T5	44	32.29	2.30	27.61	37.78	
5 - 5	T0	70	37.77	2.97	31.16	43.96	
 	T2	63	36.76	3.09	30.36	43.49	
 	T5	53	36.82	3.16	30.11	43.44	
6 - 6	T0	75	41.90	3.21	36.49	49.77	
 	T2	66	42.22	3.45	36.06	51.07	
 	T5	56	42.15	3.30	35.34	51.01	
7 - 7	T0	73	48.56	3.36	40.29	58.02	
 	T2	64	49.05	3.46	39.67	58.87	
 	T5	55	48.99	3.76	39.32	57.88	

Table 2 Changes of the mandibular intercanine, first and second premolar, and first and second molar distances over the five-year post-treatment period (in mm).

Changes of the mandibular intercanine (3-3), inter first-premolar (4-4), inter second premolar (5-5), first molar (6-6), and second molar (7-7) distances during the first two years after treatment (T0-T2), from two to five years after treatment (T2-T5), and for the whole five-year period after treatment (T0-T5) (mean, SD in mm) (paired t-test, p < 0.05).

Mandibular
distance	Time point	N	Mean	SD	Minimum	Maximum	95% confidence interval	p-value	
							Lower	Upper		
3 - 3	T0 –T2	65	0.03	0.61	−0.99	3.27	−0.12	0.18	0.698	
 	T2 –T5	50	0.05	0.44	−0.82	1.15	−0.07	0.18	0.388	
 	T0 –T5	57	0.07	0.72	−1.12	3.61	−0.12	0.26	0.444	
4 - 4	T0 –T2	49	0.65	1.09	−2.03	3.38	0.34	0.96	<0.001	
 	T2 –T5	38	0.22	0.49	−1.14	1.54	0.06	0.38	0.007	
 	T0 –T5	44	0.71	1.38	−2.12	4.61	0.29	1.12	0.001	
5 - 5	T0 –T2	63	0.92	1.37	−2.30	4.29	0.58	1.27	<0.001	
 	T2 –T5	48	0.23	0.67	−1.76	1.66	0.04	0.42	0.021	
 	T0 –T5	52	0.95	1.74	−3.70	4.70	0.47	1.43	<0.001	
6 - 6	T0 –T2	66	−0.17	1.18	−4.19	2.66	−0.46	0.12	0.247	
 	T2 –T5	50	0.05	0.62	−1.21	1.81	−0.13	0.22	0.597	
 	T0 –T5	56	−0.43	1.72	−5.34	2.71	−0.89	0.03	0.064	
7 - 7	T0 –T2	64	−0.35	1.15	−3.50	3.19	−0.63	−0.06	0.017	
 	T2 –T5	48	−0.20	1.06	−3.22	3.04	−0.51	0.10	0.188	
 	T0 –T5	54	−0.73	1.49	−4.13	2.40	−1.13	−0.33	0.001	

Transverse dental arch relationships

Table 3 shows means and SD for the MHB scores at the three different time points and Table 4 shows the changes after treatment over the five-year study period. For all arch segments –labial and cleft and non-cleft buccal segment—the MHB scores deteriorated significantly over the 5-year follow-up period and during each post-treatment period. The decrease for the labial segment was small (0.56 ± 1.45 MHB point, p = 0.006, 95% CI [0.17–0.96]) during the 5-year period, while the change for the buccal cleft segment was the largest (2.07 ± 2.67 MHB point, p < 0001, 95% CI [1.35–2.79]).

Table 3 Modified Huddart-Bodenham scores for the labial, cleft and non-cleft buccal segments and total arch constriction score directly after treatment, two and five years after treatment (in points).

Mean and standard deviation (SD) for the Modified Huddart-Bodenham scores for the labial, cleft, and non-cleft segments and total arch constriction score at the end of treatment (T0), two years after treatment (T2), and five years after treatment (T5) (in points).

Segment	Time
point	N	Mean	SD	Minimum	Maximum	
Labial	T0	73	0.49	1.92	−6	2	
	T2	64	0.33	1.73	−6	2	
	T5	56	0.09	1.74	−6	2	
Cleft	T0	73	−0.74	2.47	−8	4	
	T2	64	−1.95	2.88	−11	4	
	T5	56	−2.54	2.88	−11	2	
Non-cleft	T0	73	0.56	1.99	−6	4	
	T2	64	−0.33	1.70	−7	3	
	T5	56	−0.70	1.95	−10	2	
Total arch	T0	73	0.32	4.92	−17	8	
	T2	64	−1.95	4.86	−17	6	
 	T5	56	−3.14	4.85	−18	5	

Table 4 Deterioration of the Modified Huddart-Bodenham scores for the labial, cleft and noncleft buccal segments and total arch constriction over the five-year post-treatment period (in points).

Deterioration of the Modified Huddart-Bodenham scores for the labial, cleft and non-cleft buccal segments and total arch constriction during the first two years after treatment (T0-T2), from two to five years after treatment (T2-T5), for the whole five-year period after treatment (T0-T5) (mean, SD, in points) (paired t-test, p < 0.05).

Segment	Time point	N	Mean	SD	Minimum	Maximum	95% confidence interval	p-value	
							Lower	Upper	
Labial	T0 –T2	63	0.37	1.11	−2	4	0.09	0.65	0.011	
 	T2 –T5	51	0.24	0.74	−2	3	0.03	0.44	0.027	
 	T0 –T5	55	0.56	1.45	−4	4	0.17	0.96	0.006	
Cleft	T0 –T2	63	1.62	2.23	−3	9	1.06	2.18	<0.001	
 	T2 –T5	51	0.47	1.10	−1	4	0.16	0.78	0.004	
 	T0 –T5	55	2.07	2.67	−2	10	1.35	2.79	<0.001	
Non-cleft	T0 –T2	63	0.94	1.51	−2	4	0.56	1.32	<0.001	
 	T2 –T5	51	0.47	0.86	−1	3	0.23	0.71	<0.001	
 	T0 –T5	55	1.38	1.67	−3	6	0.93	1.83	<0.001	
Total	T0 –T2	63	2.92	3.48	−2	13	2.04	3.80	<0.001	
 	T2 –T5	51	1.18	1.83	−2	7	0.66	1.69	<0.001	
 	T0 –T5	55	4.02	4.38	−9	17	2.83	5.20	<0.001	

Factors that influence changes of mandibular transverse distances

Table 5 shows the results of the linear regression analysis for factors affecting the changes of the mandibular transverse distances for all time intervals. The deterioration of the intercanine distance two years after treatment (T0-T2) was affected by sex and GOSLON score at the end of treatment, showing that deterioration was less for girls and more when a higher GOSLON score was present at the end of treatment (R2 = 0.21). The deterioration of the inter first-premolar distance was larger when the space was closed for a missing maxillary lateral (R2 = 0.15).

Table 5 Influencing factors on decreases of the mandibular intercanine, inter first and second premolar, and inter first and second molar distances over the five-year post-treatment period.

Influencing factors (age, sex, maxillary expansion before bone grafting, space closure of the missing maxillary lateral incisor, absence of mandibular premolars, and GOSLON score after treatment (T0) on decreases of mandibular intercanine (D33), inter first-premolar (D44), inter second-premolar (D55), inter first-molar (D66), inter second-molar (D77) distances during the first two years after treatment (T0-T2), from two to five years after treatment (T2-T5), and for the whole five-year period after treatment(T0-T5) (Linear regression test).

Decreases of mandibular distances	T0 –T2	T2 –T5	T0 –T5	
	D33	D44	D55	D66	 D77	D33	D44	D55	D66	 D77	D33	D44	D55	D66	D77		
(R2)	0.21	0.15	0.07	0.07	0.15	0.06	0.06	0.09	0.14	0.25	0.13	0.08	0.05	0.08	0.14		
Age	 				 					–	–						
Sex (girl=0, boy=1)	−				 					 							
Maxillary expansion (yes=0, no=1)	 				 					 	+						
Maxillary expansion (yes=0, unknown=1)	 				 					+							
Space closure
(no=0, yes=1)	 	+	+	–	+					 							
Premolar absence (no=0, yes=1)	 				–					+							
GOSLON at T0	+	 
 	 
 	 
 	 
 	 
 	 
 	 
 	−	–	 
 	 
 	 
 	 
 	 
 		
Notes.

+ : 0.05 ≤p ≤ 0.2 with positive effect; + : p < 0.05 with positive effect.

- : 0.05 ≤p ≤ 0.2 with negative effect; − : p < 0.05 with negative effect.

Between two and five years after treatment (T2-T5), the decrease of the mandibular inter first molar width was affected by the GOSLON score at T0 showing that the decrease was less when the GOSLON at T0 was smaller (R2 = 0.14).

For the whole 5-year period (T0-T5) only maxillary expansion had a significant effect, i.e., the intercanine distance decreased more when maxillary expansion was performed during treatment, but the explained variance was low (R2 = 0.13).

Yet overall, no clear picture was emerging when looking at the effect of the different factors on the changes of the distances.

Factors that influence changes in the MHB Scores

Table 6 shows the results of the linear regression analysis for contributing factors for changes of the MHB-scores. The influencing factors had the greatest effect the first two years after treatment. From T0-T2 deterioration of the labial segment was affected by mandibular premolar absence showing that the deterioration was less if the mandibular premolar was absent (R2 = 0.17). On the buccal cleft segment, the deterioration was higher if the patient was younger at the end of treatment, space closure of a missing maxillary lateral incisor was conducted, and the GOSLON score at the end of treatment was higher (R2 = 0.31). On the buccal non-cleft segment, the deterioration was affected by patient’s age at the end of treatment and space closure for a missing maxillary lateral, showing that less deterioration happened if the patient was older at the end of treatment and no space closure was conducted for a missing maxillary lateral (R2 = 0.21). The deterioration of the total arch constriction score was smaller when the patient was older at the end of treatment, no space closure was conducted for a missing maxillary lateral and the GOSLON score was smaller at the end of treatment (R2 = 0.29).

Table 6 Influencing factors on the Modified Huddart-Bodenham scores of the labial, buccal cleft and non-cleft segment, and total arch constriction over the five-year period posttreatment.

Influencing factors (age, sex, maxillary expansion before bone grafting, space closure of missing maxillary lateral incisor, absence of mandibular premolars, GOSLON score after treatment (T0) on the Modified Huddart-Bodenham scores of labial (Lab), buccal cleft segment (CS) and non-cleft segment (NCS), and total arch constriction (Total) during the first two years after treatment (T0-T2), from two to five years after treatment (T2-T5), and for the whole five-year period after treatment (T0-T5) (Linear regression test).

Deterioration of MHB scores	T0 –T2	T2 –T5	T0 –T5	
	Lab	CS	NCS	Total	Lab	CS	NCS	Total	Lab	CS	NCS	Total	
(R2)	0.17	0.31	0.21	0.29	0.10	0.13	0.15	0.20	0.12	0.30	0.21	0.26	
Age	 	−	−	−	 				 	−	−	−	
Sex (girl=0, boy1)	 				 				 				
Maxillary expansion (yes=0, no=1)	–				 		–	–	–		–	–	
Maxillary expansion (yes=0, unknown=1)	 				−				 				
Space closure
(no=0, yes=1)	 	+	+	+	 				 	+	+	+	
Premolar absence (no=0, yes=1)	−				 	+	+	+	 	+			
GOSLON at T0	 
 	+	  	+	 
 	 
 	 
 	 
 	 
 	+	+	+	
Notes.

+ : 0.05 ≤p ≤ 0.2 with positive effect; + : p < 0.05 with positive effect.

- : 0.05 ≤p ≤ 0.2 with negative effect; − : p < 0.05 with negative effect.

Between two and five years after treatment (T2-T5), the deterioration on the buccal non-cleft segment was larger if the mandibular premolar was absent (R2 = 0.15) and the total score was more deteriorated if maxillary expansion was conducted (R2 = 0.20). For the total post-treatment period (T0-T5), age had an effect on the scores of the buccal cleft segment and the total arch constriction. The scores deteriorated less if the patients were older at the end of treatment (R2 = 0.30 and R2 = 0.26, respectively).

Overall, the effect of different factors on the deterioration of interarch relationships remained unclear because all of the explained variances were low.

Discussion

The aim of this research was to determine how the mandibular dental arch develops in patients with CUCLAP from the point the treatment has finished to five years after the treatment. We considered mandibular changes because it is well-known that the maxillary arch in patients with CUCLAP is prone to changes after treatment, whereas mandibular dental arch changes or adaptation to occlusion may be expected. Only a few studies have been performed on long-term changes after treatment in adult patients with CUCLAP and we found none focusing on the transverse dental arch dimensions in the mandible. Therefore, comparison with findings from the literature is not possible as data is lacking. The null hypothesis could not be confirmed as the present study shows that the mandibular arch does change after the treatment has finished as shown by changes of the transverse dimensions and interarch relationship over the 5-year follow-up period.

The mandibular intercanine distance did not change significantly over time as could be expected because 85.3% of the patients (64/75) had a mandibular canine-to-canine retainer bonded to all anterior teeth. The interpremolar distances diminished significantly over the 5-year study period with about 1 mm. The first and second molar distance increased slightly, but this was only significant for the second molars (0.73 mm). The latter can be explained by uprighting of the mandibular molars buccally from a more lingually oriented crown torque which occurs with age in untreated dentitions leading to an increase of the intermolar distance (Marshall et al., 2003; Hesby et al., 2006; Yang & Chung, 2019). However, it should be noticed that changes of the intermolar distance were minor which is consistent with earlier longitudinal studies (Thilander, 2009; Heikinheimo et al., 2012; Garib et al., 2021).

In our study the interpremolar distances diminished slightly while studies on untreated subjects have shown that interpremolar distances in the mandible reached stability between 16 and 31 years of age (Thilander, 2009) and remained stable from 13 to 60 years of age (Massaro et al., 2018). These changes in the interpremolar distances might be explained as an adaptation to changes in the maxillary dental arch in CUCLAP which is subject to the ongoing constrictive effect of palatal scar tissue as has been shown in animal experiments in which animals were followed into adulthood (Wijdeveld et al., 1991; Kim et al., 2002; Van De Water, Varney & Tomasek, 2013).

In contrast to the lack of data on mandibular arch dimensions in patients with orofacial clefts there are a few studies that reported posttreatment changes of the occlusion and maxillary arch dimensions. Marcusson & Paulin (2004) evaluated occlusion and maxillary arch dimensions from 19 to 25 years of age with the MHB Index and found that the deteriorations were significant in all three segments. Semb, Ronning & Abyholm (2011) did a long-term study on antero-posterior relationship using the GOSLON Yardstick and observed that from 16 to 20 years of age in 30% of patients the GOSLON score worsened. The changes were partly due to continuous mandibular growth. A series of studies from Brazil in a convenience UCLP sample reported on short term changes (one year after treatment) primarily of the maxillary dental arch in relation to rehabilitation with a dental implant or anterior fixed bridgework (Caballero et al., 2019; Rezende Pucciarelli et al., 2020; Pucciarelli et al., 2021; Soares et al., 2021a; Soares et al., 2021b). These studies showed that the maxillary arch was not stable until one year posttreatment regardless of the type of prosthetic rehabilitation. In our research, the MHB scores deteriorated up to 4 points for the total score which means there is an increase in the number of teeth in crossbite. The interarch relationship deteriorated and maxillary arch changes in the long term may be responsible for the increased MHB scores. Yet the mandibular arch showed only minor changes of the transverse dimensions. It means that the mandibular arch did not adapt completely to the changes in the maxillary arch dimensions because if it did the MHB scores would have remained the same.

We also studied factors which may influence the changes of the mandibular arch and the MHB score after treatment. We did not find a clear pattern for the effect of age at the end of treatment, sex, maxillary expansion before bone grafting, space closure for missing maxillary lateral, mandibular premolar absence or GOSLON score at the end of treatment, on the mandibular transverse dimensions over the 5-year follow-up period. For the interarch relationship we found that age at the end of treatment, space closure for a missing maxillary lateral incisor and GOSLON score at the end of treatment seemed to have an effect especially in the first two years after treatment. Patients who were younger at the end of treatment tended to have more changes, probably because of remaining growth. Indeed, it has been reported that patients with clefts would mature slower than non-clefts due to delayed pubertal growth (Cesur et al., 2018). Therefore, it should be taken into account that there may be a risk of post-treatment changes if we finish treatment while growth has not ceased. We also found that the post-treatment changes were less if patients had a lower GOSLON score at the end of treatment, which stands for a better sagittal jaw relationship. This is encouraging to know because it helps us to inform the patients what to expect of their treatment stability.

This study shows that changes after treatment occurred especially during the first two years after treatment. Therefore, clinicians should anticipate with their retention protocol. Bonded retainers to all anterior teeth are recommended to hold the anterior tooth positions. It is recommended to use, besides the bonded retainers, a Hawley retainer during the night to maintain the transverse dimensions after treatment. Due to the long-term effect of palatal scar tissue the patient must be advised to use the Hawley retainer life-long. It was also shown that patients who had a good GOSLON score at the end of treatment maintained a better transverse dental arch relationship overtime. Therefore, it pays off to strive for the lowest GOSLON score at the end of treatment, although this is not always possible due to the individual growth pattern of the patient.

A common feature in patients with CUCLAP is agenesis of the maxillary lateral incisor at the cleft side. In this study we found that if the space of missing maxillary lateral was closed the interarch deterioration in the first two years after treatment was larger compared to opening the space for a lateral incisor. This may be explained by the fact that closing the space narrows the maxillary dental arch facilitating a cross bite tendency which will be reflected in deterioration of the MHB index. This raises the question whether it would be wise to open the space and replace the missing maxillary lateral (temporarily) with a Maryland bridge or dental implant. A single dental implant in the anterior region can only be placed after vertical facial growth has ceased (Kuijpers & Loomans, 2015; Aarts et al., 2015). A noticeable infra-occlusion may also occur due to continuous vertical eruption of the adjacent teeth even though implant placement is done at mature age (Bernard et al., 2004). Furthermore, in many cases additional bone grafting is necessary before dental implant placement because bone volume loss occurs after secondary bone grafting when no tooth has erupted or placed orthodontically in the area (Stasiak, Wojtaszek-Slomińska & Racka-Pilszak, 2019; Wermker et al., 2014). Also, peri-implant soft tissues are esthetically less pleasing in patients who did not receive additional bone augmentation as the implant cannot be placed in its optimal position due to an insufficient bone volume (Alberga et al. (2020). However, so far there are no long-term studies of dental implants in patients with clefts comparing therapies, risks, and outcome (Wermker et al., 2014; Wang et al., 2014).

There is still a lack of studies into long-term changes in patients with CUCLAP not only regarding the dental arch relationship and facial changes but also regarding other aspects of treatment outcome. The adult individual with a treated cleft deserves more attention in cleft research.

Strengths and weaknesses of the study

This study is performed in a large sample of consecutive patients observed for 5 years after treatment. There are hardly any studies about stability after treatment focusing on the mandibular arch. The loss to follow up was low (only 7 out of 82 patients—less than 10%). The patients were treated from birth on in one center with the same protocols by two experienced surgeons which is important to notice as it has been proven that the skills of the surgeon play an important role in the final result (Shaw & Semb, 2017). However, the present study is a retrospective study and performed in one center which means the results are valid for Caucasians and for the treatment protocol which is employed in the center.

Conclusions

In patients with CUCLAP, changes occurred in the mandibular arch expressed as changes in the transverse dimensions and interarch relationship measured by the MHB Index. A younger age at the end of treatment, space closure for a missing maxillary lateral incisor and a higher GOSLON score at the end of treatment negatively influence the interarch transverse deterioration especially in the first two years after treatment. For the transverse dimensional changes in the mandibular arch such influencing factors could not be determined.

Supplemental Information

Supplemental Information 1 Intra-observer reliability and measurement error for the mandibular transverse distances

Intra-observer reliability and measurement error for the mandibular intercanine (3-3,) inter first premolar (4-4), inter second premolar (5-5), inter first molar (6-6), inter second molar (7-7) distances at 3 different time points. Reliability expressed by Pearson’s correlation coefficient. DME = duplicate measurement error (in mm); Mean diff = mean difference between first and second measurement and 95% confidence interval (in mm). Results of paired t-test for the mean diff (p-values).

Click here for additional data file.

Supplemental Information 2 Bland Altman plots of mandibular width at T0

Bland Altman plots of mandibular distances at the end of treatment (intra-observer).

Click here for additional data file.

Supplemental Information 3 Bland Altman plots of mandibular width at T2

Bland Altman plots of mandibular distances at 2 years after treatment (intra-observer).

Click here for additional data file.

Supplemental Information 4 Bland Altman plots of mandibular width at T5

Bland Altman plots of mandibular distances at 5 years after treatment (intra-observer).

Click here for additional data file.

Supplemental Information 5 Bland Altman plots of the Modified Huddart-Bodenham scores (intra-observer)

Bland Altman plots of the Modified Huddart-Bodenham scores (intra-observer) between measurement 1 (M1) and measurement 2 (M2) for the labial segment, buccal cleft and non-cleft segment, and total arch constriction.

Click here for additional data file.

Supplemental Information 6 Bland Altman plots of the Modified Huddart-Bodenham scores (interobserver)

Bland Altman plots of the Modified Huddart-Bodenham score (inter-observer) for the labial segment, buccal cleft and non-cleft segment, and for total arch constriction.

Click here for additional data file.

Supplemental Information 7 Raw dataset

Click here for additional data file.

We are grateful for the contributions by Lovina Boen, DDS (Radboudumc Nijmegen) and Dr. Nia Ayu Ismaniati, DDS PhD (Universitas Indonesia Jakarta) for their support in the data collection.

Additional Information and Declarations

Competing Interests

Author Contributions

Human Ethics

Data Deposition

Anne Marie Kuijpers-Jagtman is an academic editor for PeerJ. All other authors declare that they have no competing interests.

Sariesendy Sumardi performed the experiments, analyzed the data, prepared figures and/or tables, authored or reviewed drafts of the paper, and approved the final draft.

Benny S. Latief conceived and designed the experiments, prepared figures and/or tables, authored or reviewed drafts of the paper, and approved the final draft.

Anne Marie Kuijpers-Jagtman conceived and designed the experiments, analyzed the data, prepared figures and/or tables, authored or reviewed drafts of the paper, and approved the final draft.

Edwin M. Ongkosuwito performed the experiments, authored or reviewed drafts of the paper, and approved the final draft.

Ewald M. Bronkhorst analyzed the data, authored or reviewed drafts of the paper, and approved the final draft.

Mette A.R. Kuijpers conceived and designed the experiments, performed the experiments, analyzed the data, authored or reviewed drafts of the paper, and approved the final draft.

The following information was supplied relating to ethical approvals (i.e., approving body and any reference numbers):

This study has been carried out in accordance with the applicable Dutch legislation such as Medical Research involving Human Subjects Act and Medical Treatment Contracts Act concerning reviewal by an accredited research ethics committee. This has been confirmed by the Medical Ethical Committee of the Radboudumc Nijmegen, The Netherlands (File number: 2021-13046).

The following information was supplied regarding data availability:

The raw data and measurements are available in the Supplementary File.

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
