# Peer review of "Long-term follow-up of mandibular dental arch changes in patients with complete non-syndromic unilateral cleft lip, alveolus, and palate"

_PeerJ, doi:10.7717/peerj.12643_

## Round 0.1 · original submission · Minor Revisions

The reviewers have commented on several issues of your manuscript, which I think warrant some attention.

Reviewer 1 ·

Basic reporting

1. Plaster models of 75 consecutive patients directly after (T0), 2 years after (T2), and five years after treatment (T5) were available.
T0 - age?
T2 - age?
T5 - age?
The total should be 225. Why 197?
2. complete non-syndromic unilateral cleft lip, alveolus, and palate - any problems in mandible?
3. Line 111 - 130 = Details related to the maxilla. Anything with mandible?
4. GOSLON and MhB measured specially designed to check the condition of the maxilla. What about mandible?
5. 20 Models re-measured 1 wk interval. I think it is too early. Should be 2 weeks or 3 weeks.
6. Changes values are shown. Mean values followed by changes data are easy to follow.
7. Figure 1 - is it from the computer/software generated?
8. Figure 3 - adapted from?

Experimental design

1. Line 111 - 130 = Details related to the maxilla. Anything with mandible?
2. GOSLON and MhB measured specially designed to check the condition of the maxilla. What about mandible?

Validity of the findings

# 20 Models re-measured 1 wk interval. I think it is too early. Should be 2 weeks or 3 weeks.

Additional comments

None

Reviewer 2 ·

Basic reporting

Clear and unambiguous, professional English used throughout. There are a few comments to improve

Page 8, line 70 - CLP is not defined as a short form. The authors have defined terms as cleft lip alveolus and palate but not cleft lip and palate. CLP should be defined as cleft lip and palate when used for the first time. I would also recommend to use one consistent terminology, if possible, throughout the manuscript.

Line 103 - “The aim of this study was to determine long term changes in transversal mandibular … to determine interarch transverse relationship in patients with ..post-treatment.” Please use the word transverse consistently rather than using transverse and transversal interchangeably.

Line 106 “We also studied risk factors that could…” Please use third person rather than first person language such as In this study, or the risk factors that could ….were studied”

Line 117 “ Exclusion criteria were: additional craniofacial….in(complete) or submucous cleft lip at the …” I think the authors meant to convey incomplete cleft as exclusion - the brackets before and after the word complete should be removed.

Line 154 - MHB index - the full form should be written for the first time when it appears in the main text.

The terms T0, T2, and T5 are confusing. They imply that there were 5 time-points. Rather the terms T0, T1, and T2 would be more appropriate with T1 defined as 2 years and T2 defined as 5 years after treatment.

Line 297”In contrast to the lack of data on mandibular arch dimensions in patients with clefts there are a few studies that”. Please use a consistent terminology CLP rather than ‘clefts’

Line 326 - the word bigger is inappropriate rather an adjective such as higher or larger may be more appropriate.

Experimental design

The study is well conducted. The strength of the study is the long followup of patient for 5 years. There are a few comments as mentioned below:

“Line 307 “It means that the mandibular arch did not adapt completely to the maxillary arch because if it did the MHB scores would not det...” I would suggest to rephrase this sentence. Rather than the mandibular arch not adapting the maxillary arch - the MHB scores are affected due to changes in maxillary arch as mandibular arches changes were minimal. Therefore, it can be reworded as maxillary arch changes in the long term may be responsible for the increased MHB scores.


Line 328. Please consider replacing the terms upper dental arch with maxillary arch and lower dental arch with mandibular arch throughout the manuscript. Similarly, changes can be done upper to maxillary when referring to any teeth as well.

Validity of the findings

Intra and inter observer reliability have been stated. Results are adequately explained.

Additional comments

Line 346 - please add a couple lines on the clinical implications and how can use the results of this study be useful for clinical practice.

Reviewer 3 ·

Basic reporting

It was a pleasure to review this well-crafted manuscript. The English language is clear, and the literature is almost enough. I suggested some papers to add and collaborate with this relevant scientific study.

Experimental design

It is original primary research. The null hypothesis is not defined, but it does not belittle the study. There are some flawns to understand the methods, and I suggested adding to collaborate with the investigation.

Validity of the findings

It is a subject relevant and still not evaluated. The literature is well established, despite there are few studies about the theme. This investigation will contribute to knowledge.
All statistical analyses are robust, and the conclusions are well stated, supported by results.

Additional comments

Thanks for PeerJ for the indication of my name to review the paper entitled:

Long-term Follow-up of Mandibular Dental Arch Changes in Patients with Complete Non-syndromic Unilateral Cleft Lip Alveolus and Palate

I only have minor suggestions to recommend before publication to enhance the readers' comprehension of the study findings.

Abstract
Line: 39
It is relevant to consider that 75 patients, three different moments result in 225 mandibular dental casts. What happened with 28 models? Or remove the dental casts number.

Keywords
Lines 60, 61
Mandibular arch is not a MESH term.
Long term outcome is not a MESH term.
Complete unilateral cleft lip is not a MESH term.
Cleft lip is a MESH term.
Cleft palate is a MESH term.
Modified Huddart-Bodenham Index is not a MESH term.

Introduction
Lines 85, 86
I suggest read the paper:
Pucciarelli MGR, Toyoshima GH, Cardoso JF, de Oliveira TM, Neppelenbroek KH, Soares S. Arch Asymmetry in Patients With Cleft Lip and Palate After Rehabilitation Treatment Using Stereophotogrammetry. J Craniofac Surg. 2021 Jul-Aug 01;32(5):e501-e504. doi: 10.1097/SCS.0000000000007460. PMID: 33481468.

Line 105
I suggest including the null hypothesis.

Materials & Methods
Lines 120-131
Were these patients separated in raw data? For patients who needed superiorly based pharyngeal flap at 4-5 years, maxillary protraction and expansion at 5-6 years for 12-18 months followed by minor lip/ala nose correction and finally severe dysgnathia, combined orthodontic-surgical treatment followed by lip/nose correction was performed. How did the authors deal with these patients?

Line 141
Please, clarify the moment of the beginning of the study. After treatment? Which treatment? Orthodontic treatment? Orthognathic surgery?

Line 169
Which one is correct here, the signal – or AND?

Line 173
The first time that DME appears. Please, what does it mean? Inform the meaning in brackets.
DME: duplicate measurement error

Results
Lines 193-195
One patient is absent here, maybe the patient is in the group of the maxillary expansion before bone grafting, or the group of the lateral incisor space closed orthodontically.

Line 212
It is too much figure. Think to be reasonable.

Discussion
Lines 295-297
Is there any scientific evidence to support this statement?

Lines 308-309
If the mandibular arch did not adapt completely to the maxillary arch, how can the authors affirm that changes in the interpremolar distances might be explained as an adaptation to changes in the maxillary dental arch in CUCLAP which is subject to the ongoing constrictive effect of palatal scar tissue?

Lines 322-324
Maybe one of the most relevant pieces of information about the study.

Lines 330-331
Pease read these papers:
1.Pucciarelli MGR, Toyoshima GH, Cardoso JF, de Oliveira TM, Neppelenbroek KH, Soares S. Arch Asymmetry in Patients With Cleft Lip and Palate After Rehabilitation Treatment Using Stereophotogrammetry. J Craniofac Surg. 2021 Jul-Aug 01;32(5):e501-e504. doi: 10.1097/SCS.0000000000007460. PMID: 33481468.
2. Rezende Pucciarelli MG, de Lima Toyoshima GH, Marchini Oliveira T, Marques Honório H, Sforza C, Soares S. Assessment of dental arch stability after orthodontic treatment and oral rehabilitation in complete unilateral cleft lip and palate and non-clefts patients using 3D stereophotogrammetry. BMC Oral Health. 2020 May 27;20(1):154. doi: 10.1186/s12903-020-01143-1. PMID: 32460814; PMCID: PMC7254638.
3. Soares S, Rezende Pucciarelli MG, Hideki de Lima Toyoshima G, Marchini Oliveira T. Stereophotogrammetry to evaluate young adults with and without cleft lip and palate after orthodontic and restorative treatment. J Prosthet Dent. 2021 Feb 12:S0022-3913(20)30733-2. doi: 10.1016/j.prosdent.2020.10.025. Epub ahead of print. PMID: 33589235.

Line 352
Please read this paper:
Caballero JT, Pucciarelli MGR, Pazmiño VFC, Curvêllo VP, Menezes M, Sforza C, Soares S. 3D comparison of dental arch stability in patients with and without cleft lip and palate after orthodontic/rehabilitative treatment. J Appl Oral Sci. 2019 Jun 13;27:e20180434. doi: 10.1590/1678-7757-2018-0434. PMID: 31215598; PMCID: PMC6559757.

Annotated reviews are not available for download in order to protect the identity of reviewers who chose to remain anonymous.

---

## Round 0.2 · accepted · Accept

Many thanks for this exceptional submission to the Journal.

Reviewer 2 ·

Basic reporting

NA

Experimental design

NA

Validity of the findings

NA

Additional comments

The authors have done appropriate modifications

Reviewer 3 ·

Basic reporting

See attached PDF for corrections.

Experimental design

No comment.

Validity of the findings

No comment.

Additional comments

Please, correct the reference: Pucciarelli MGR, Toyoshima GH, Cardoso JF, de Oliveira TM, Neppelenbroek KH, Soares S. Arch Asymmetry in Patients With Cleft Lip and Palate After Rehabilitation Treatment Using Stereophotogrammetry. J Craniofac Surg. 2021 Jul-Aug 01;32(5):e501-e504. doi: 10.1097/SCS.0000000000007460. PMID: 33481468.

Annotated reviews are not available for download in order to protect the identity of reviewers who chose to remain anonymous.